# Integration of Sentiment Analysis of Social Media in the Strategic Planning Process to Generate the Balanced Scorecard

José Roberto Grande-Ramírez, Eduardo Roldán-Reyes *, Alberto A. Aguilar-Lasserre [ID] and Ulises Juárez-Martínez

National Technology of Mexico TecNM, Campus Orizaba, Oriente 9 #852, Col. Emiliano Zapata, Veracruz 94320, Mexico
* Correspondence: eduardo.rr@orizaba.tecnm.mx

**Abstract:** Strategic planning (SP) requires attention and constant updating and is a crucial process for guaranteeing the efficient performance of companies. This article proposes a novel approach applied in a case study whereby a balanced scorecard (BSC) was generated that integrated sentiment analysis (SA) of social media (SM) and took advantage of the valuable knowledge of these sources. In this study, opinions were consolidated in the main dataset to incorporate sentiments regarding the strategic part of a restaurant in a tourist city. The proposed methodology began with the selection of the company. Information was then acquired to apply pre-processing, processing, evaluation, and validation that is capitalized in a BSC to support strategic decision-making. Python support was used in the model and comprised lexicon and machine learning approaches for the SA. The significant knowledge in the comments was automatically oriented toward the key performance indicators (KPIs) and perspectives of a BSC that were previously determined by a group of opinion leaders of the company. The methods, techniques, and algorithms of SA and SP showed that unstructured textual information can be processed and capitalized efficiently for optimal management and decision-making. The results revealed an improvement (reduced effort and time) to produce a more robust and comprehensive BSC with the support and validation of experts. Moreover, new resources and approaches were developed to implement more efficient SP. The model was based on the efficient coupling of both fields of study.

**Keywords:** strategic planning; sentiment analysis; social media; key performance indicators; balanced scorecard

## 1. Introduction

Business performance management (BPM) enables organizations to maximize their performance and achieve their aims through a set of information systems, business processes, and technological tools that focus on strategic indicators [1,2]. As a fundamental part of this organizational culture, strategic planning (SP) allows the establishment of necessary elements to determine the guidance of an organization [3]. According to George et al. [4], SP is one of five management approaches that companies should consider. The actual value for the successful execution of SP depends on several factors. Nevertheless, shared leadership inspires all company collaborators to carry out SP and encourage real change by achieving its objectives [5].

Balanced scorecard (BSC) has proven to be an effective management mechanism for measuring strategic performance [6]. This method is structured based on four perspectives: finance, customers, internal processes, and learning and growth. The determination of key performance indicators (KPIs) focuses on monitoring performance against an organization's objectives. As a reinforcement of its classic elaboration, we consider that external sources, such as social media (SM) can contribute to its automatic development from a large part of its perspective. The fundamental reason is that today, it is impossible to talk about an organization without considering SM as a corporate strategy. The key to business success

for many companies is the correct use of data to make faster and better decisions and to reduce failures [7].

Currently, a company's relationships with its customers are generated in real-time. The large amount of SM data generated daily offers enterprises new opportunities. Understanding their influence on customers or users through their comments is essential [8,9]. Globally, businesses collect data on consumer preferences, purchases, and trends. The correct interpretation of sentiment in SM allows for the prediction of market behavior and can incur profits or losses within companies [10].

In addition, SA analyzes people's opinions and sentiment intention and evaluates and appraises the emotions people post about a product, service, organization, topic, individual, or trend [11]. The basis of SM and opinion mining (OM) models consist of algorithmic techniques [12].

SA faces general stages, such as defining data related to the topics, data collection from a source, sentiment classification, polarity detection, presenting sentiment information in a good summary, and finally, making the decision. The methodology proposed in this study aimed to be straightforward, easy to apply, and efficient. As a first step, the company was selected to guarantee the application. Then, the information from social networks was acquired, to which the pre-processing, processing, evaluation, and validation that determines the KPIs of interest were applied and capitalized in a BSC to support strategic decision-making. This system will make it possible to clarify the initiatives that must be implemented to improve the business. SA allows automation and reduces information processing time during the SP process. The contribution of this study is that the SA of SM can be considered for SP with satisfactory results. Consequently, this study arose from a literature review where an area of opportunity was detected in relation to generating a BSC by integrating SA with SP. Significant classic efforts have been made in business strategy. Still, the gap we are trying to fill is part of generating a BSC fed from external sources (SM), and providing a more complete and robust mechanism to support strategic decision-making focused on its four perspectives.

From the above, the following research question (RQ) arose: Do the opinions of SM allow strengthening and adding value to a BSC for better strategic decision-making?

This paper is comprised of six sections. Section 2 describes work related to this proposed approach, which is almost zero. In Section 3, the proposed methodology is explained clearly and precisely. Section 4 presents the implementation of the system in a real industry case. This is then discussed in Section 5. Finally, the conclusions are summarized in Section 6.

## 2. Related Work

### 2.1. Strategic Planning and Balanced Scorecard

SP is an essential and latent process for a system, sector, or organization. Several studies have been directly related to this topic, and the different applications and approaches they can generate. In addition to the classic business process, SP has been applied in various fields, such as the health sector [13–16]. Strategic recommendations have recently been generated owing to the COVID-19 pandemic [17]. Moreover, it has been considered in the logistics of small and medium-sized enterprises (SMEs) [18], the introduction of new products in the automotive industry [19], and maintenance [20].

George et al. [4] published an article that generated a random-effects meta-analysis, which suggested that SP positively and significantly affected organizational performance. In the same study, a meta-regression analysis showed that SP on organizational performance is more positive when the performance is measured as effectiveness and when SP is measured as formal strategic planning. The link between the latter concept and planning flexibility is positively related to a company's capacity for innovation [21]. The approach applied by Papke-Shields and Boyer-Wright [22] confirmed that the application characteristics of SP in project management can be beneficial. In the public sector, statistical findings are presented using SEM-partial least squares, where there is a positive relationship be-

tween SP, strategic implementation, excellence, and organizational performance [23]. The promotion of SP and the alignment of educational objectives are essential for improving the performance of public institutions [24].

As part of SP, the BSC is considered an effective mechanism for monitoring performance. In research developed by Craig and Moores [25], the authors recommend the BSC, with its four perspectives on how family-owned businesses can develop and manage a succession plan, to maintain the essence of the founders. Another scientific work proposed an approach integrated with the BSC and knowledge-based system (KBS) by applying the analytical hierarchy method (AHP); the results indicate that this approach facilitates efficient automated strategic planning [26]. A novel study used an optimization system based on a genetic algorithm to determine the optimal organizational objectives for the conformation of a BSC [27]. Furthermore, ERP (enterprise resources planning) systems can prove their strategic contribution through a BSC, as suggested by Chand et al. [28]. Similarly, the BSC has demonstrated its strategic effectiveness in education [29], decision-making for energy investment [30], improving business evaluation using various approaches [31], and other fields of study.

### 2.2. Sentiment Analysis of Social Media

In recent years, multiple contributions from the SA of SM have been generated because of the valuable knowledge that can be acquired. Customer reviews of one or several products have been an invaluable source for improving their design and features [32–35], understanding the sentiment of opinions, and enhancing the service of restaurants [36,37], hotels [38,39], and the tourism industry in general [40,41], and even for the improvement of airport services [42]. Because of its wide variety and utility, SA has been used to predict and forecast markets, as shown by Kraaijeveld and De Smedt [43], and Zaidi and Oussalah [44]. SA applications in marketing and sales have been highlighted [45–50]. Complementing the scope of the SA application, essential contributions of the method have also been made in the health sector, as has been recently applied in several research studies [51–55].

### 2.3. Strategic Planning and Sentiment Analysis

The integration of SP and SA presents a gap in the literature. Although scientific research has been conducted on these topics, there remains an important area of opportunity. Srinivas and Rajendran [56] analyzed the online reviews of students for the strategic planning of universities. This approach identifies strengths and weaknesses from students' perspectives and detects opportunities and threats from competitors' perspectives. The results indicated an improvement in the recruitment and retention of students in schools. Similarly to previous work, Na et al. [57] analyzed CEO messages in management reports to predict corporate financial ratios reflected in six suggested perspectives of a sustainability-balanced scorecard (SBSC). Another article by Krishnamoorthy [58] used financial and non-financial KPIs to analyze the sentiment of investors, institutions, and markets from economic texts. One study indicated that communication strategies improve customer relationship management through the SA of SM [59]. In an article by Kazmaier and van Vuuren [60], an interesting strategic proposal is shown that takes advantage of SA to support organizations to inform decision-making effectively. This combines unstructured and structured data. Recently, Ahmadi and Qaisari Hasan Abadi [61] highlighted that the generation of strategic plans takes a long time and is expensive, so they proposed an expert SP system that reduces the above in two steps: first, the system is discerned, and second, the indicators are prioritized.

On the other hand, Kurnia [62] created a business intelligence model that observes the topic of interest in SM and supports decision-making for beneficial purposes for companies. Finally, Vasquez-Rojas et al. [63] used an approach integrating text mining with SP. Their objective was to create efficient strategic plans that do not require significant resources for SMEs. These results are significant because they can be obtained in the application of a case study.

Related studies have recently emerged about strategic decision-making based on SA on particular issues. A study by Elbarachi et al. [64] analyzed comments regarding climate change and concluded that the geographical location, the culture of a country, and the political influence of a president are influential factors in the formation of public opinion. Consequently, the strategy regarding some topics of interest may vary due to these circumstances. Jianjun et al. [65] proposed methods to define and identify the critical opinions of online consumers that companies could prioritize to optimize their online response strategies while incorporating the latest artificial intelligence technology to handle a torrential volume of information. Regarding online marketing strategies, Micu et al. [66] verified that SA can be used to interpret customer behavior and highlight how pre-sale, sale, and post-sale strategies can be improved. Similarly, Huang et al. [67] designed a three-stage framework for strategic marketing planning, incorporating multiple benefits of AI: mechanical AI to automate repetitive marketing functions and activities, thinking AI to process data to reach decisions, and sensitive AI to analyze human interactions and emotions.

In the health sector, a study by Acosta et al. [68] inferred that applying NLP techniques in a smaller proportion of data through transfer learning was useful for obtaining sufficient precision in the sentiment analysis stages. The study focused on improving care in times of COVID-19 to improve the generation of strategies. Within the same sector, Shah et al. [69] proposed a novel method for OM that integrated artificial intelligence and semantic web techniques to analyze opinions in order to assess patients' perceptions about the quality of the service of medical care. The findings contribute to better overall performance analysis for healthcare organizations. The work of Song et al. [70] generated a sustainable corporate governance strategy based on the sentiment analysis of financial reports with corporate social responsibility. The study supported investors in making better decisions. Finally, Venugopal et al. [71] proposed a SA model for the challenges of strategic financial management in emerging companies. The results showed greater efficiency in SA for new companies.

The novel proposal of this article contributes to state-of-the-art research by incorporating three concepts (SA of SM with SP). The result was the determination of KPIs and automatic generation of the BSC in a case study. This contribution supports company improvements in control and performance.

## 3. Research Methodology

The first step of the proposed approach was to select a company and its sector. Subsequently, the datasets were selected and generated. Python 3.7 was used to perform most of the system. In the sentiment analysis process, we pre-processed the information, which involved the stages of feature selection and extraction, sentiment classification, polarity detection, and evaluation and validation. Subsequently, we created an automatic database to access the final report and analysis that included the KPIs and perspectives of the BSC.

Finally, a case study was presented in which the investigation was applied, and where the company relied on the study to make better strategic decisions to ensure it operation and growth. Figure 1 illustrates the proposed research methodology.

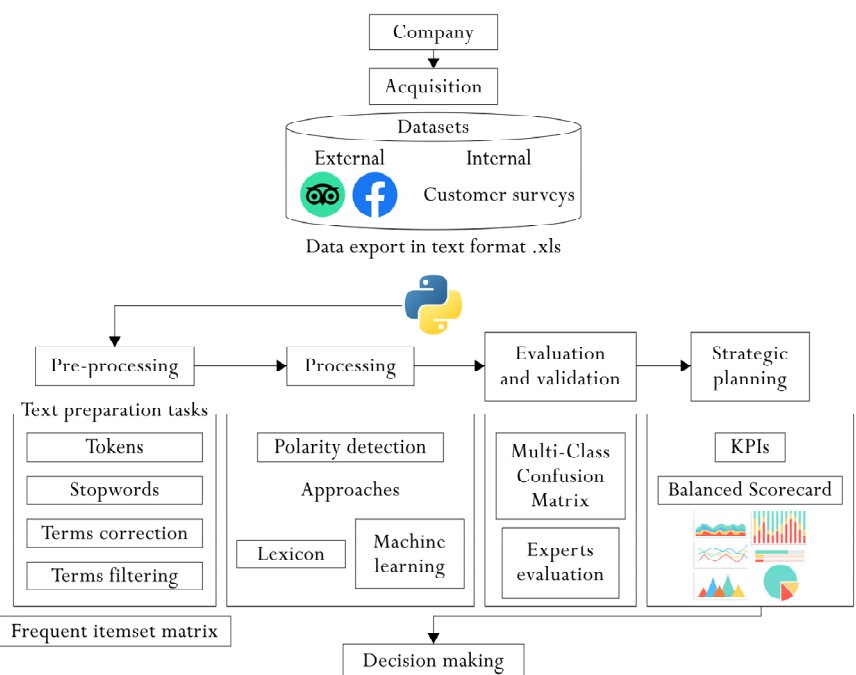

**Figure 1.** Research methodology.

### 3.1. Datasets

Various sources were reviewed to determine which were the most appropriate for the study. In the process, which social media platform had the largest number of followers and opinions about the business were reviewed. After checking Twitter, Instagram, YouTube, Facebook, and TripAdvisor, it was determined that the last two were the most consistent in the established criteria. Surveys were administered to the most frequent customers to visualize an internal reference. The collected data were exported and stored in spreadsheet format.

#### 3.1.1. Why TripAdvisor and Facebook?

TripAdvisor is the world's largest travel platform and helps users find the best options based on opinions and comments about the experiences of other users. This website is an ideal dataset for SA research because of the millions of reviews about hotels, restaurants, airlines, tourist attractions, and so on. Comments contain a minimum of 200 characters and users can rank their reviews using a numerical score [72]. TripAdvisor includes interactive travel forums [73].

Facebook is the most popular social media platform and makes it simple to share photos, text messages, videos, news, status posts, and sentiments. As of January 2020, Facebook had approximately 2500 million active users per month; 61% of internet users use this platform. An average user accesses Facebook eight times daily, followed by Instagram with six, and Twitter with five. The platform generates approximately 400 new users per minute.

The growth of active companies and individual users has allowed Facebook to become a company/brand for customer interactions [59]. The company reported over 2.31 billion active accounts in early 2019. The concept of "groups" or "communities" has contributed to many users because several people can interact to share content based on a common theme. The group concept is helpful for OM research, and posts tend to be related to the articles adopted by the group [54]. In this era of SM, information flows quickly through social media platforms, such as Facebook, and is mainly negative comments about products or services [45]. There are documented cases of Facebook live streams of people who receive a defective product or lousy service.

### 3.1.2. Internal Survey

However, a substantial percentage of customers do not always express their opinions on SM. Therefore, many enterprises have decided to consult their customers' experiences through email, forums, face-to-face surveys, mobile devices, applications, or Google forms. This practice is helpful because the customer's voice provides valuable information to find opportunities for improvement.

### 3.2. Pre-Processing

The collected information must be refined and transformed gradually through different sub-processes.

Case conversion. In most sentiment analysis cases, lowercase words do not exactly match uppercase words and vice versa. The solution to this problem was to convert the input words into a single-case format, that is, uppercase or lowercase, to facilitate post-processing [63].

Tokenization. At this stage, opinions are fragmented into syntactic processing units called tokens (i.e., words) or groups of tokens called n-grams. The conversion of text into a set of tokens was shown after removing punctuation marks, accents, or hyphens [51].

Stop-word filtering. This process eliminates words that do not provide helpful information and generates noise during the process. Common stop-words include prepositions, pronouns, articles, or conjunctions.

Terms correction. At this stage, the opinion volume can be reduced by up to 12 opinion volumes. Each token was compared using a formal local language dictionary. If a term did not exist, the token was removed.

Terms filtering. Filtering reduces text dimensions. In this case, the comments were compared to the vocabulary of the text domain. This step simplified the process.

After the pre-processing stage, the opinion base reduced its dimensionality by up to 40%, obtaining the most helpful information for the next step.

### 3.3. Processing

In this step, information was processed using different tools and approaches to extract implicit knowledge and transform it into explicit knowledge.

Polarity Determination

The polarity of a sentence determines its orientation in a positive, negative, or neutral manner. It is commonly used in the reviews of blogs, microblogs, or forums. The type and amount of information are primarily unstructured and require high-level processing and intelligent analytical techniques. There are four approaches to this task: semantic, machine learning, deep learning, and hybrid. The methods used in this study were as follows.

Lexicon-based approach (LBA): LBA depends on the sentiment lexicon and uses a bank of pre-coded words to determine the semantic orientation of the text [74]. This method includes two approaches: first, a dictionary is created with some words as a base; then, the antonyms and synonyms of those words are searched and new words are added to improve it. The corpus-based approach determines sentiment orientation from a context-specific list of words. This approach comprises two sub-approaches: semantic and statistical [75]. LBA is widely used in conventional texts, such as reviews, forums, and blogs. However, it is less likely to be used for big data extracted from SM websites [76]. LBS has the disadvantage of offering unacceptable accuracy rates because the polarity shown in the lexicon may distort polarity. This situation can be improved by generating context-specific lexicons [77,78]. LBA does not require training data (as the supervised machine-learning method does) and has been used in previous studies to identify sentiments, particularly for Twitter and Sina Weibo.

The valence-aware dictionary for sentiment reasoning (VADER): This is a rule-based lexical sentiment analysis tool that adapts to text analysis to generate a score (polarity). It uses modified scores for lexical attributes present in a sentence and determines the overall

average sentiment score. The particular design of this method is suitable for the analysis of social networks; however, it can analyze texts of any length [79]. Moreover, it generates an intensity to determine whether a sentiment is positive or negative. VADER has better precision than other types of linguistic analyses [80]. This is one reason why it is an ideal tool for our research.

Machine Learning Approach (MLA): Machine learning (ML) has been developed in recent years and offers new ways to address real-world challenges. The MLA uses linguistic methods and applies common ML algorithms. These are classified into supervised and unsupervised approaches. Supervised learning methods aim to determine the desired output variable for a given input variable. The algorithm is identified using a learning method through mapping. In an unsupervised manner, there is no adequate input data. It aims to model the underlying structure or distribution to learn more about the data and identify the output.

Supervised learning algorithms require training data for sentiment classifier learning, are domain-dependent, and require retraining with the arrival of new data. The learning stops when the algorithm reaches an acceptable level of performance.

In MLA, classifiers are primarily trained using a feature set composed of n-grams, and there is a need for a large number of known examples to train them. Studies have shown that these types of classifiers perform better than lexicon-based ones. This situation is limited to a single domain and is non-transferable between applications. This limitation is one reason researchers have introduced "hybrid approaches" [81].

According to the literature, support vector machine (SVM) and Naive Bayes (NB) are the most commonly used classifiers [55,82–84]. The SVM is a classification algorithm that aims to define the most significant margin of separation of the hyperplane of classes [37]. This algorithm is efficient and suitable for use with textual data. According to Salas-Zárate et al. [85], there are three main advantages of this classifier. First, it is robust in high-dimensional spaces and when there is a sparse set of samples. Second, any type of feature is relevant. Third, many text-categorization problems are linearly separable. However, it can be challenging to interpret the results outside of their accuracy values because the SVM assumes no linearity.

On the other hand, NB is the most popular algorithm for classifying documents. It has a high speed and accuracy when used in large databases. NB has statistical and probability bases and predicts future values based on experience. The advantage of this method is that it requires a small amount of training data to determine the parameters required for classification [86]. One limitation of NB is the assumption of the independent predictors.

### 3.4. Evaluation and Validation

Multi-Class Confusion Matrix

The confusion matrix allows the visualization of the performance of an algorithm. The evaluation of this model presents two variants: quantitative and qualitative. The first is estimated by calculating the precision of the frequency at which a document is correctly categorized. Three data classes were considered in this study: positive, negative, and neutral. The accuracy is represented by the number of correctly classified data instances divided by the total number of instances. This metric is not very reliable when there is a high skew in the classes; therefore, we also strengthened it with precision, recall, and F—measure [87]. Precision is the percentage of documents correctly classified and recall represents the percentage of papers categorized by the classifier. By contrast, the F-measure harmonically balances both precision and recall [88]:

$$\text{Accuracy} = \frac{\text{TP} + \text{TN}}{\text{TP} + \text{FP} + \text{FN} + \text{TN}} \tag{1}$$

$$\text{Precision} = \frac{\text{TP}}{\text{TP} + \text{FP}} \tag{2}$$

$$\text{Recall} = \frac{\text{TP}}{\text{TP} + \text{FN}} \tag{3}$$

$$\text{F} - \text{measure} = \frac{2 * \text{Precision} * \text{Recall}}{\text{Precision} + \text{Recall}} \tag{4}$$

where true positives (TP) are correctly classified, true negatives (TN) refer to the number of instances where the classifier accurately predicts the negative class as unfavorable, false positives (FP) are the number of predictions where the classifier incorrectly predicts the negative class as positive, and false negatives (FN) are the number of instances where the classifier incorrectly predicts the positive type as negative.

Finally, during the qualitative stage of the research, a group of experts manually labeled the comments to validate the polarity. This strategy allowed for the incorporation of valuable knowledge from experts, capitalizing on more robust and reliable work.

### 3.5. Strategic Planning

Involvement, leadership, dialogue, and a shared vision are necessary to successfully implement an SP in an organization [5]. Once the SA was developed, the SP was incorporated. The procedure was performed using an affinity diagram that allowed the classification and presentation of the information of interest in an orderly manner. The process adopted in this study was novel because it was performed automatically using Python. First, from all the opinions of the consolidated dataset, a word cloud with the highest polarity and frequency was created. Second, using the similarity index and Jaccard distance, the most representative words for each opinion were oriented toward the appropriate KPI.

Jaccard Distance and Similarity Index

The Jaccard similarity index was chosen for this study because of its simplicity, ease of use, and effectiveness of its results. This method has been applied in various fields of study with satisfactory results. Jaccard similarity was defined as the ratio of the size of the intersection to the size of the union of sample sets [89]. Equation (5) represents this as follows:

$$\text{Sim} \, (A, B) \, \text{Jaccard} = \frac{|R_A \cap R_B|}{|R_A \cup R_B|} = \frac{|R_A \cap R_B|}{|R_A| + |R_B| - |R_A \cap R_B|} \tag{5}$$

where $R_A$ is the set of tokens present in the customer comment, and $R_B$ is the set of tokens present in the KPI. $R_A$ and $R_B$ are mutually exclusive; therefore, the addition rule theorem applies. Simultaneously, the Jaccard distance score measures the dissimilarity between the clustering results [90] and is calculated using Equation (6):

$$\text{Jaccard distance} = 1 - \text{Sim} \, (A, B) \, \text{Jaccard} \tag{6}$$

When KPIs have already been selected, they are categorized according to each perspective that makes up the enterprise's BSC. This is shown in Figure 2. Finally, this proposal will support strategic decisions that will ensure operations, market growth, the introduction of new products, and other strategies.

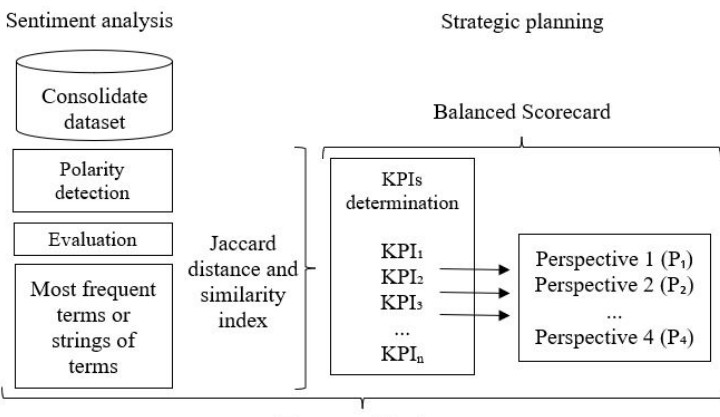

**Figure 2.** Automatic integration of sentiment analysis and strategic planning.

## 4. Application: Case Study

The proposed model was applied to Madison Grill, a roast meat restaurant in Orizaba, Veracruz, Mexico (www.gob.mx/sectur/articulos/orizaba-veracruz, accessed on 18 February 2022). This branch was part of a chain of restaurants in the country. The city is one of 132 magical towns (www.orizaba.travel/, accessed on 18 February 2022) in the national territory, whose attractions generate great admiration among national and foreign tourists (www.visitmexico.com/en/veracruz/orizaba, accessed on 20 February 2022). The local government had consolidated tourist attractions; therefore, companies in the restaurant sector saw a need to continuously improve their processes and services.

An exploratory analysis of social networks revealed that the tourist customers of Madison Grill wrote reviews on TripAdvisor (www.tripadvisor.com.mx/ShowUserReviews-g108 2256-d8505941-r641895357-Madison_Grill-Orizaba_Central_Mexico_and_Gulf_Coast.html, accessed on 10 March 2022). Similarly, the restaurant's Facebook (www.facebook.com/ MadisonGrillOrizaba/reviews/?ref=page_internal, accessed on 23 March 2022) page had active users and a significant number of comments for performing SA. In contrast, the planning team collected opinions from frequent customers. The latter did not express an opinion on SM to avoid duplicating reviews or creating noise in sentiment. Table 1 lists the characteristics of the datasets used in this study.

**Table 1.** Case study databases.

| Dataset | Data Collection Period | Languages | Extraction Mode | Reviews | % |
|---|---|---|---|---|---|
| Tripadvisor | 2017–February 2022 | Spanish, English, Italian, German, French | API | 155 | 23 |
| Facebook | 2017–February 2022 | Spanish | API | 269 | 39 |
| Customer Survey | 2019–January 2022 | Spanish, English | Manual (Tablet) | 261 | 38 |
| | | | Total | 685 | 100 |

The collected data were exported to a .xls file, generating the primary dataset.

### 4.1. Pre-Processing

In the pre-processing stage, we used the Natural Language Toolkit (NLTK) in Python 3.7. NLTK is an industrial-strength platform that has a wide range of natural language processing libraries. This method was used, and the stages described in Section 2.2 were carried out: (1) case conversion, (2) tokenization, (3) stop-word filtering, (4) terms correction, and (5) terms filtering or non-significant words. These stages were followed to prepare for all the comments. Table 2 shows a TripAdvisor review that exemplifies pre-processing.

**Table 2.** Pre-processing steps of a real opinion.

| Original Opinion | English Translation |
| --- | --- |
| 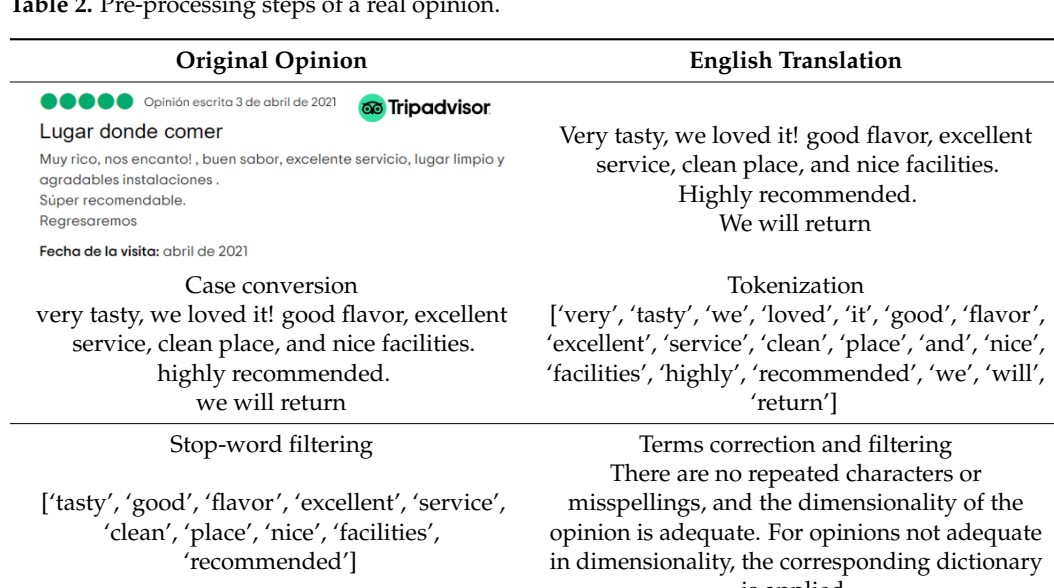 | Very tasty, we loved it! good flavor, excellent service, clean place, and nice facilities. Highly recommended. We will return |
| Case conversion<br>very tasty, we loved it! good flavor, excellent service, clean place, and nice facilities. highly recommended. we will return | Tokenization<br>['very', 'tasty', 'we', 'loved', 'it', 'good', 'flavor', 'excellent', 'service', 'clean', 'place', 'and', 'nice', 'facilities', 'highly', 'recommended', 'we', 'will', 'return'] |
| Stop-word filtering<br><br>['tasty', 'good', 'flavor', 'excellent', 'service', 'clean', 'place', 'nice', 'facilities', 'recommended'] | Terms correction and filtering<br>There are no repeated characters or misspellings, and the dimensionality of the opinion is adequate. For opinions not adequate in dimensionality, the corresponding dictionary is applied. |

Figure 3 shows the percentage reduction after each pre-processing task.

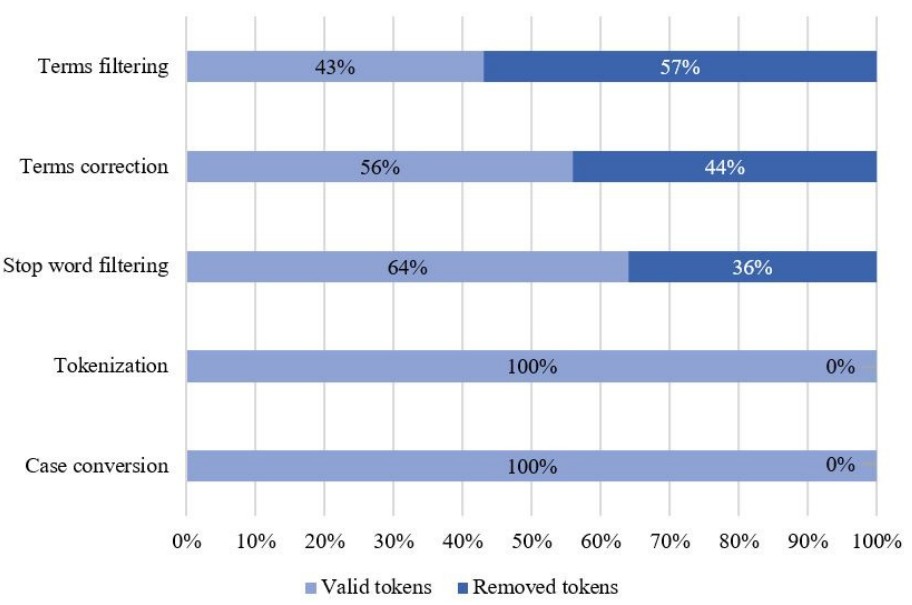

**Figure 3.** Valid tokens proportion after pre-processing.

The token reduction was approximately 40%. The resulting tokens were grouped into a frequent itemset matrix, as shown in Table 3. The rows indicate each of the 685 instances of the opinion, the columns represent the 2620 valid tokens in the model, and the cells represent the frequencies of the terms. The cells assume values directly proportional to the number of tokens counted, which are represented as $\{Tk_i, Tk_2, Tk_3, \ldots, Tk_n\}$ for each of the textual instances, which are represented as $\{Ti_1, Ti_2, Ti_3, \ldots, Ti_n\}$. If this was not satisfied, the value was automatically set to 0.

**Table 3.** Frequent itemset matrix.

| Instance IDs | $Tk_1$ | $Tk_2$ | $Tk_3$ | $Tk_4$ | $Tk_5...$ | $Tk_{2620}$ |
|---|---|---|---|---|---|---|
| $Ti_1$ | 0 | 0 | 0 | 1 | 0 | 0 |
| $Ti_2$ | 0 | 0 | 0 | 0 | 0 | 0 |
| $Ti_3$ | 0 | 0 | 0 | 0 | 0 | 1 |
| $Ti_4$ | 1 | 0 | 0 | 0 | 0 | 0 |
| $Ti_5...$ | ... | ... | ... | ... | ... | ... |
| $Ti_{685}...$ | 0 | 0 | 1 | 0 | 0 | 0 |

*4.2. Processing*

This step generates the polarity calculation. Two approaches were used: LBA using VADER [91], and MLA using SVM and NB classifiers. In the first approach, the installation was performed with the line of code > pip install vaderSentiment and imported with the vaderSentiment import SentimentIntensityAnalyzer. The values were normalized for polarity between $[-1,1]$ [60] and a word was determined as positive if the related value was greater than 0.05, as negative if the negative value was less than $-0.05$, and as neutral if the value was greater than or equal to $-0.05$ and less than or equal to 0.05. The overall polarity of opinions was calculated by summing the determined polarities. Regarding the classification, the program generated a general polarity label. SVM and NB were used for the machine learning approach. The results of both methods are shown in Figure 4.

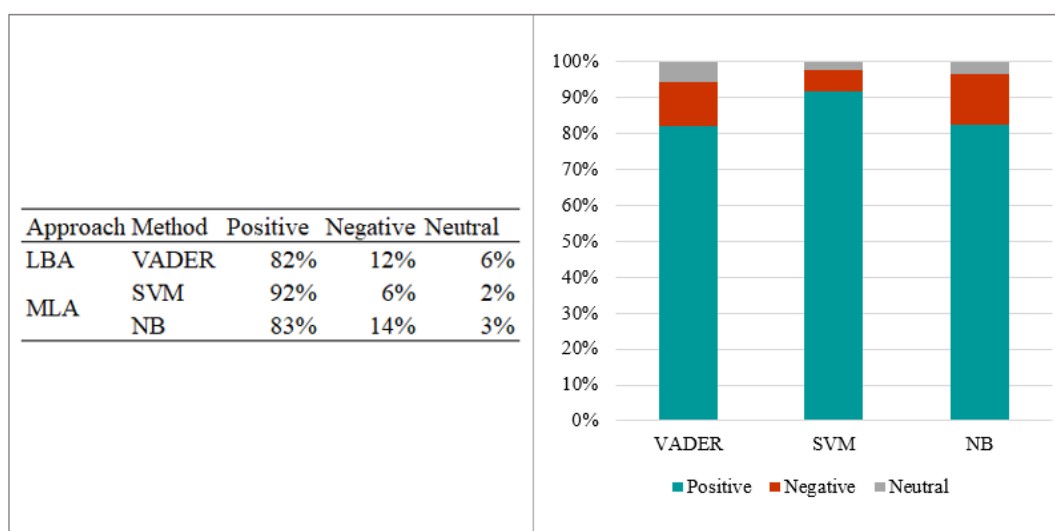

**Figure 4.** Proportion of opinion polarity by approach.

It can be observed that in the two approaches and three methods implemented, opinions were positively oriented toward the restaurant. In addition, the negative percentage of polarity ranged from 6% to 14%. Neutrality was present in the opinions to a minor degree.

*4.3. Evaluation and Validation*

In the evaluation stage of the LBA, we used a manual method similar to that proposed by D'Andrea et al. [51], which consists of randomly selecting and labeling training comments. The difference is that in this work, we involved members of the SP group to classify the comments as positive, negative, or neutral. The test set consisted of labeling all comments from the primary dataset. For LBA, Figure 5a shows a multi-class confusion matrix containing the classification results for the test dataset. The results were obtained using Python programming language. The entries are shaded according to the number of observations in each category, with the largest number of observations is represented by the darker shade. It can be determined that the VADER model was "confounded" mainly by negative and neutral observations.

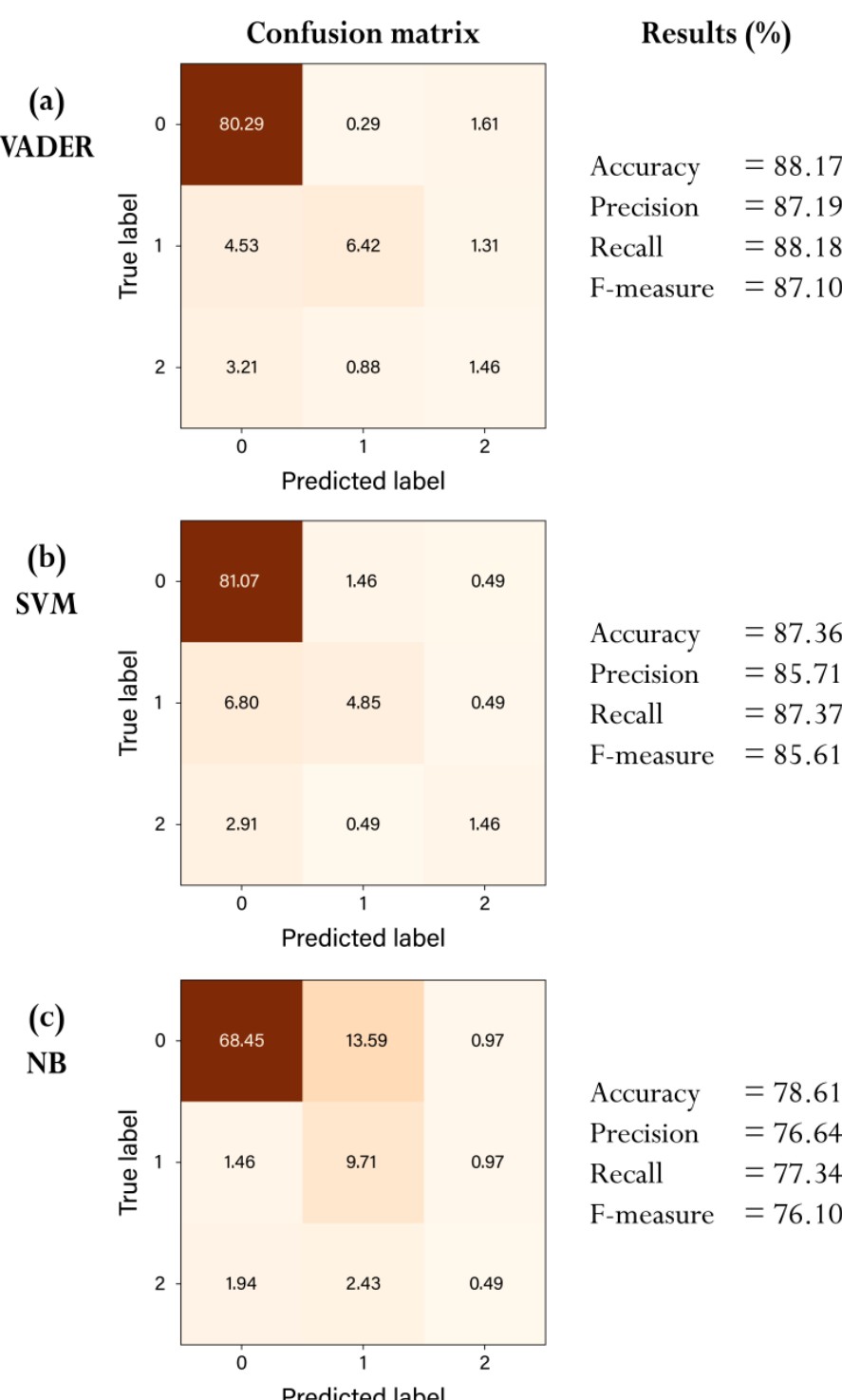

**Figure 5.** Confusion matrices and evaluation metrics of the VADER, SVM, and NB methods.

Figure 5b,c show the representations used to evaluate SVM and NB within the MLA. The dataset was divided into training and testing sets using 5-Fold cross-validation. The GridSearchCV method was used to determine suitable SVM parameters. This tuning technique attempts to calculate the optimal values of the hyperparameters and is called tuning. The results showed C: 14.0 and gamma: 0.0625. It can be seen that the results of the evaluation metrics were better for VADER, SVM, and then NB. Hutto and Gilbert [91] found that VADER performed better for specific datasets. The representation of the normalized matrix permits the visualization of models that perform better with the metrics used [60].

Finally, the planning group was followed up. This model was deemed more reliable and robust because it considered expertise.

### 4.4. Strategic Planning

During the SP process for the Madison Grill restaurant, a team was integrated with opinion leaders who were responsible for transmitting the actions that must be applied to achieve the enterprise's objectives. Implementing the strategic stage involves the OM of SM because the amount of information, its diversity, and the richness of the content are necessary attributes to formulating a strategic plan, and in this case, a BSC. The classic strategy formulation processes involve manual mechanisms that consume considerable time and effort [92]. This situation presents an automatic system that operates using unstructured information in a short time.

After processing, the framework presented a word cloud (tokens) with the highest polarity and frequency in SA (Figure 6).

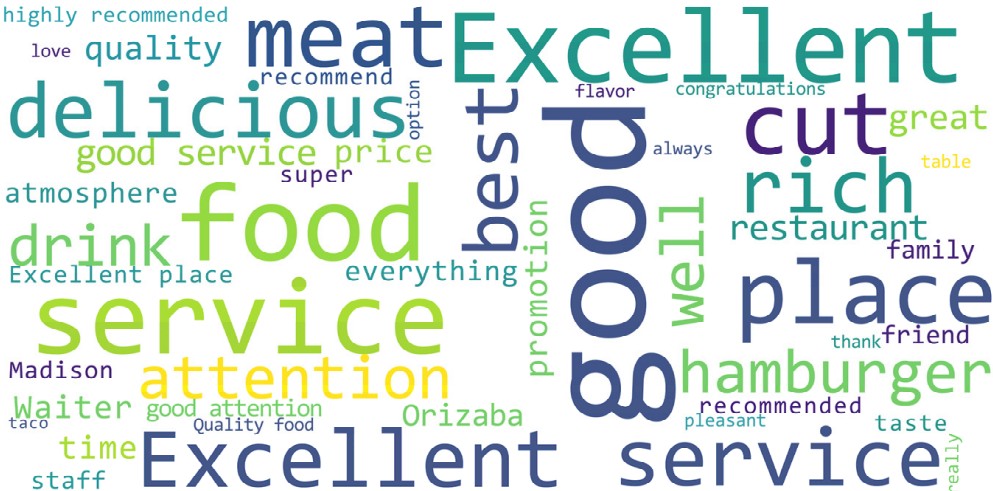

**Figure 6.** A word cloud illustrating the most frequent tokens, the final processed dataset.

Each comment was oriented to a specific KPI of interest ($KPI_1$-$KPI_9$) previously validated by the planning team. Using Python, the Jaccard similarity index (Equation (5)) was used to determine the relationship between the intersection referring to the set of tokens present in the opinions and the set of tokens present in the KPIs. The union between the two sets was analyzed. Jaccard distance (Equation (6)), used in the Python code as jaccard_distance = 1—jaccard_similarity, was used to calculate the distance of the tokens present in the comments to the KPIs and categorize each of the BSC perspectives. The planning team validated the KPIs and their orientations to each perspective using an affinity diagram. Figure 7 shows a representation of the BSC architecture, which was automatically generated according to valuable information from customer reviews.

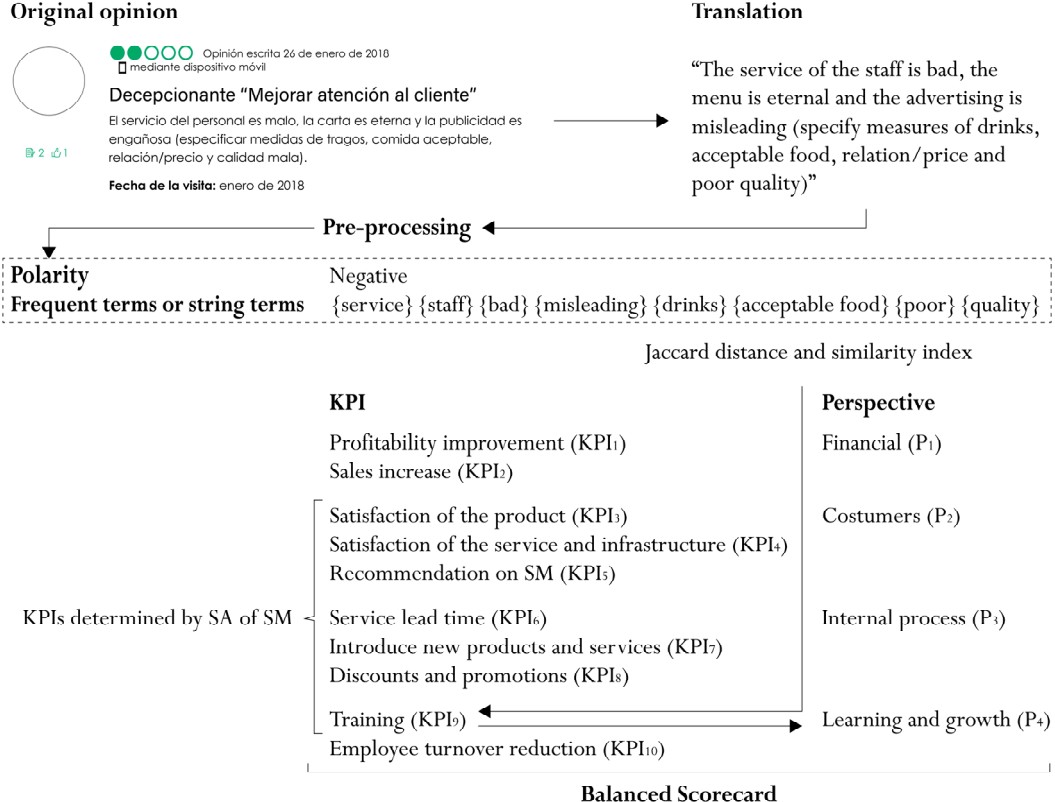

**Figure 7.** Representation of the automatic generation of the balanced scorecard.

The customers' opinions of SM were mainly oriented to perspective 2 ($P_2$-Costumers), which automatically generated KPIs 3, 4, and 5. The planning team should consider this perspective as a priority. The reason for this is that the customers do not necessarily know either the financial status or the internal process of the restaurant. Still, their opinions were considered an alternative for visualization in monitoring the indicators. It is essential to mention that the SP team manually incorporated the financial perspective ($P_1$-$KPI_1$ and $KPI_2$) and the learning and growth perspective ($P_4$-$KPI_9$ and $KPI_{10}$) by the SP team with the aim of creating the complete BSC. To complete the above, it was necessary to follow up with experts in order to validate the results. Table 4 shows the percentage of opinions categorized by KPIs and perspective.

**Table 4.** Percentage of opinions categorized in the KPIs and perspectives.

| Perspective | % Comments | KPI | % Comments |
|:---:|:---:|:---|:---:|
| ($P_2$) | 74.6% | $KPI_3$ | 12.0% |
| | | $KPI_4$ | 45.1% |
| | | $KPI_5$ | 17.5% |
| | | $KPI_6$ | 7.3% |
| ($P_3$) | 12.8% | $KPI_7$ | 1.8% |
| | | $KPI_8$ | 3.8% |
| ($P_4$) | 10.7% | $KPI_9$ | 10.7% |
| | | N/A | 1.9% |
| | | Total | 100.0% |

It can be seen that the customer perspective was the one with the highest categorization of comments, mainly in $KPI_4$, which indicates that attention should not be lost in this category since the customer tends to like or dislike it to a greater extent. Based on the automatic configuration of the restaurant's BSC, the proposed system was considered adequate for the company because it was aligned with its mission, vision, and objectives.

## 5. Discussion

The proposed model demonstrated technical efficiency through a case study of a restaurant in a tourist city. The results showed that SA of SM can be successfully incorporated into PE, is relevant for generating a BSC, and offers support for making better strategic decisions. In addition, the system is versatile and can be applied to various companies regardless of the size and type of business. To reinforce the implementation, we suggest that the planning team commit to the importance of SP and value potentially useful information that can be obtained from the opinions of SM customers. Natural language processing (NPL) continues to challenge researchers, so the method used here aimed to correct spelling errors, abbreviations, spams, etc. The application of various approaches, such as LBA and MLA, allowed for the comparison of polarity results and contributed to the algorithms employed. Python programming language and the support of its libraries enabled the process to be made more efficient, quicker, and automatic in the model. This work allows for the incorporation of technological tools and innovation into an enterprise's processes. As a medium-term goal, it is intended to completely digitize processes and develop marketable software to assist the SP task in this sector. The main advantages of the proposed model concerning related work are as follows:

- Incorporating two lines of research that seem opposed but work together, that is, the SA of SM in SP for the generation of a BSC.
- The system offers the opportunity to integrate and recognize expert knowledge and continuously validate the results.
- The application of opinion mining algorithms enables the detection of customer sentiments and their satisfactory grouping towards the KPIs of interest.
- The information can be treated anonymously without violating the rights of the SM users or discriminating against the opinions of the planning team.
- The automatic processing of this information reduces the time required for this type of exercise by approximately 70%.

The proposed model demonstrates that SP can be addressed and improved using several methods and approaches. Despite the benefits highlighted by this study, more effort is required to address some drawbacks. For instance, on the one hand, within the existing limitations in the model, we understand that, at the moment, there is no friendly and easy-to-use interface. In the same way, programming knowledge is required so that the system can be powered. On the other hand, the databases used should be expanded so that the results have greater consistency.

In future work, it is proposed that the system can detect the bots that may exist in the comments issued, the influence behind the post, and the sarcasm within the opinions. Moreover, the opportunity to integrate more robust and consistent databases, possibly from new SM, is becoming increasingly important for companies. The recent announcement of Meta by Facebook has evolved the digital connection of SM. Therefore, we consider that more powerful methods, such as ours, will contribute to the evolution and benefits of what this implies.

## 6. Conclusions

In this study, the proposed model automatically generated a BSC after performing a SA of SM. This process was validated using expert knowledge. The results were satisfactory and demonstrated in a case study. The contribution to the state of the art is important because the literature that incorporates SA and SP is scarce. This approach aimed to determine that KPIs that are most appropriate for businesses and can guide them to the perspectives of the BSC.

In contrast to the classic construction of this process, it was also decided to integrate valuable SM knowledge to strengthen the BSC tool. The implementation of efficient approaches and sophisticated methods highlights the advantages of saving time and improving manual processing.



The results of the generated BSC show that in Mexico, an evolution focused on improving services has begun. Thus, emerging economies generally prioritize product quality. As Mexican society is aimed at becoming a developed country, becoming involved in professionalizing services and showing a culture of significant improvement is essential. This case study makes it possible to promote a strategic planning culture that involves all employees, which implies an awareness and recognition of SP as a critical process in this type of business.

We consider our model to be technically suitable and meet the versatility required by any organization. Finally, knowledge and human experience are not replaced. Instead, the model is a mechanism that can make the process more efficient and help stakeholders to make efficient strategic decisions to ensure and grow a company.

**Author Contributions:** Conceptualization, methodology, formal analysis, and investigation, J.R.G.-R. and E.R.-R.; software, J.R.G.-R.; writing—original draft preparation, J.R.G.-R.; writing—review and editing, J.R.G.-R., E.R.-R. and A.A.A.-L.; visualization, supervision, and validation, J.R.G.-R., A.A.A.-L. and U.J.-M. All authors have read and agreed to the published version of the manuscript.

**Funding:** This research was funded by the National Council of Science and Technology of Mexico (CONACyT) through the scholarship grant: CVU 344910, and the National Technology of Mexico, Campus Orizaba.

**Institutional Review Board Statement:** Not applicable.

**Informed Consent Statement:** Not applicable.

**Conflicts of Interest:** The authors declare no conflict of interest.

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
