# Peer review of "Integration of Sentiment Analysis of Social Media in the Strategic Planning Process to Generate the Balanced Scorecard"

_applsci, doi:10.3390/app122312307_

Round 1

Reviewer 1 Report

The paper proposes to generate the balanced scorecard with the help of sentiment analysis of the social media. The authors suggest to use the sentiment/emotion analysis from social media for assessing the KPIs of the BSC. We find the research idea is novel and applicable. The proposed approach is illustrated using a case study on a restaurant.Generally, the paper is clear and well-written.

We only regret about the use of the social media to measure the KPIs of the four perspectives of the BSC. It is valid to use the social media to detect the customer emotions towards a specified brand or product. Consequently, the social media can be used to predict the customer satisfaction KPIs. This satisfaction could be related to the service provided or a product. But, it is not valid the use the social media to identify the KPIs of the internal process, financial or even the learning and growth. The internal process is not known to the customers, as well the financial status of the organization is not known to the customers. As performed in the case study, you identified only the customer satisfaction about the provided service.  Therefore, we suggest to revise the manuscript to measure only the KPIs of the customer perspectives. Only measuring the KPIs of the customer perspective is an appreciated contribution.

Reviewer 2 Report

The topic is very interesting, proposing a novel approach applied in a case study in which a balanced scorecard is generated that integrates sentiment analysis of social media.

The paper is written in good English and is easy to read. All the methods are described clearly and precisely.

Some of the picures are not in very good quality and need improvement.

Reviewer 3 Report

Thank you for the opportunity to review this paper. The authors have written an interesting piece but I have a few comments:

First, the introduction does a good job highlighting the importance of the balance scored card. However, I do not see much about what has been already investigated in relation to the topic of the paper and what is the research gap the authors try to fill.

Similarly, it would be good if the authors can include a clear research question (in the form of question, ending in a question mark).

I would also recommend to expand the section 2.3 about sentiment, since it is a core construct in your paper but it only has one paragraph in the literature review.

I think it would be important to specify the methodology chosen briefly in the abstract and in the introduction, to inform the reader about what the paper actually does.

In section 3.1 you mention “After exhaustive review of various sources…” Please be specific as this kind of statements are a bit unclear to readers.

Please describe the limitations of the study and the potential avenues for further research that arise from your work.

Please note that the fond of the text is not homogenous throughout the manuscript.

Good luck!

Round 2

Reviewer 3 Report

I have no further comments